# Quantifying plasmid movement in drug-resistant *Shigella* species using phylodynamic inference

Nicola F. Müller[1,2]*, Ryan R. Wick[3], Louise M. Judd[4], Deborah A. Williamson[5,6], Trevor Bedford[2,7], Benjamin P. Howden[3,4], Sebastián Duchêne[3,8,9]☯*, Danielle J. Ingle[3,4]☯*

**1** Division of HIV, ID and Global Medicine, University of California San Francisco, San Francisco, California, United States of America, **2** Vaccine and Infectious Disease Division, Fred Hutchinson Cancer Center, Seattle, Washington State, United States of America, **3** Department of Microbiology and Immunology at the Peter Doherty Institute for Infection and Immunity, The University of Melbourne, Melbourne, Victoria, Australia, **4** Center for Pathogen Genomics, The University of Melbourne, Melbourne, Victoria, Australia, **5** Department of Infectious Diseases at the Peter Doherty Institute for Infection and Immunity, The University of Melbourne, Melbourne, Victoria, Australia, **6** School of Medicine, University of St Andrews, Fife, Scotland, **7** Howard Hughes Medical Institute, Seattle, Washington State, United States of America, **8** EDID Unit, Department of Computational Biology, Institut Pasteur, Paris, France, **9** Université Paris Cité, Paris, France

☯ These authors contributed equally to this work.
* nicola.felix.mueller@gmail.com (NFM); sebastian.duchene@pasteur.fr (SD); danielle.ingle@unimelb.edu.au (DJI)

## Abstract

The 'silent pandemic' of antimicrobial resistance (AMR) represents a significant global public health threat. AMR genes in bacteria are often carried on mobile elements, such as plasmids. The horizontal movement of plasmids allows AMR genes and resistance to key therapeutics to disseminate in a population. However, the quantification of the movement of plasmids remains challenging with existing computational approaches. Here, we introduce a novel method that allows us to reconstruct and quantify the movement of plasmids in bacterial populations over time. To do so, we model chromosomal and plasmid DNA co-evolution using a joint coalescent and plasmid transfer process in a Bayesian phylogenetic network approach. This approach reconstructs differences in the evolutionary history of plasmids and chromosomes to reconstruct instances where plasmids likely move between bacterial lineages while accounting for parameter uncertainty. We apply this new approach to a five-year dataset of *Shigella*, exploring the plasmid transfer rates of five different plasmids with different AMR and virulence profiles. In doing so, we reconstruct the co-evolution of the large *Shigella* virulence plasmid with the chromosome DNA. We quantify higher plasmid transfer rates of three small plasmids that move between lineages of *Shigella sonnei*. Finally, we determine the recent dissemination of a multidrug-resistant plasmid between *S. sonnei* and *S. flexneri* lineages in multiple independent events and through steady growth in prevalence since 2010.

**Data availability statement:** The source code for the analyses performed, such as the R scripts to recreate figures, is available at https://github.com/nicfel/Plasmids-Material. Details of the isolates included in this study are available in S1 and S2 Tables. The coalescent with plasmid transfer is implemented as a package to BEAST2 called CoalPT. The source code for this package is available here https://github.com/nicfel/CoalPT.

**Funding:** N.F.M. was supported in part by NIH NIGMS R35 GM119774. S.D. is supported by the Inception program (Investissement d'Avenir grant ANR-16-CONV-0005). R.R.W. is supported by the Australian Research Council (ARC) through a Discovery Early Career Researcher Award (DECRA) [DE250100677]. T.B. is an Investigator of the Howard Hughes Medical Institute. B.P.H. is supported by an NHMRC Investigator Grant (GNT1196103). D.J.I. is supported by an NHMRC Investigator Grant (GNT1195210). This work was supported by a National Health and Medical Research Council (NHMRC) Australia partnership grant (GNT1149991). The funders had no role in the study design, data collection and analysis, decision to publish, or preparation of the manuscript.

**Competing interests:** The authors have declared that no competing interests exist.

This approach has a strong potential to improve our understanding of the evolutionary dynamics of AMR-carrying plasmids as they are introduced, circulate, and are maintained in bacterial populations.

## Author summary

Bacterial pathogens can evolve antimicrobial resistance (AMR) either via mutations or the acquisition of AMR genes by horizontal gene transfer on mobile genetic elements, such as plasmids. While the former can be studied with existing evolutionary tools, the latter is challenging with current methods. This manuscript introduces a novel method that jointly reconstructs the horizontal movement of AMR genes from the discordance of plasmid and chromosomal DNA. This method enables us to quantify the frequency with which plasmids move between bacterial lineages. We demonstrate the utility of this approach on AMR and virulence plasmids in *Shigella*, quantifying different transfer rates within and between *Shigella* species. This novel approach will be applicable to the investigation of plasmid dynamics in bacterial populations.

## Introduction

Antimicrobial resistance (AMR) in bacteria represents one of the most serious public health threats of the 21$^{st}$ century [1–3]. Bacterial pathogens can evolve AMR either through mutations in core genes or via the acquisition of AMR genes by horizontal gene transfer on mobile genetic elements, such as plasmids. The horizontal transfer of genetic material, including AMR genes, allows bacterial populations to rapidly adapt and evolve under changing selective pressures and ecological niches [4–6]. Plasmids are typically considered to be part of the accessory genome; genes that are variably present across genomes that are drawn from the species pangenome [7–9]. Some plasmids can move between lineages of the same bacterial species or between unrelated bacterial species [8]. Multiple studies to date have identified plasmids as playing central roles in driving the emergence, spread, and increasing prevalence of AMR in several bacterial species, including *Klebsiella pneumoniae*, *Salmonella enterica*, *Escherichia coli*, and *Shigella* species [5,10–13]. Quantifying the horizontal movement of plasmids in populations and modeling rates of transfer of plasmids in bacterial species where drug resistance is often driven by plasmids is a major barrier to our understanding of the drivers of the prevalence and dissemination of AMR. Hence, there is an unmet need for novel computational approaches to quantify the movement and spread of mobile elements in the accessory genome of bacterial pathogens [3].

*Shigella* is an exemplar bacterial pathogen to explore the methodological approaches to quantify the evolution and movement of plasmids. There are four species of *Shigella,* with two species, *Shigella sonnei* and *Shigella flexneri*, responsible for the global burden of shigellosis [14–17]. *Shigella* is a WHO AMR priority pathogen due to increased resistance to clinical therapeutics [2]. *Shigella* has

been shown to carry different plasmids that vary in size and function (AMR, virulence), and with different evolutionary histories [12–14,16]. While the pINV plasmid is considered part of the core genome, plasmid content varies within and between the *Shigella* species. For example, in *S. sonnei*, three other smaller plasmids have been characterized in the reference genome Ss046. These three small plasmids, spA, spB, and spC, are commonly found within *S. sonnei* global lineage III [16]. More recently, attention has focused on the movement and spread of a large multidrug-resistant plasmid (MDR; resistant to three or more antimicrobials), particularly in men who have sex with men (MSM) [12,13,18–21]. Importantly, these outbreaks have been driven by variants of the MDR plasmid, pKSR100, which mediates AMR to co-trimoxazole, azithromycin, and ampicillin, moving between populations of *Shigella* [12,13,19,22,23]. The horizontal transfer of plasmids occurs between bacterial lineages and, as such, is not a co-divergent process with the chromosomal DNA of these bacterial lineages.

Here, we model this process by using a novel 'coalescent with plasmid transfer' (CoalPt) model. CoalPT is an approach to reconstructing the acquisition, movement, and co-divergence of plasmids in routinely generated WGS data. Our method is implemented as a package for the open-source software BEAST2 [24] to facilitate its adoption. As such, one can use the evolutionary models already implemented in BEAST2. The plasmid transfer rate is a population-level rate and a function of how often bacterial lineages are in the same location, the probability of them exchanging plasmids (if they have one), and also the degree of selection that acts on the bacterium that picked up a new plasmid. We used empirical data from *Shigella* isolates to explore and quantify the plasmid transfer rates of several plasmids within *Shigella* that differ in size, AMR, and virulence profile, and expected evolutionary history. We then show that modeling the co-divergence of plasmid and chromosomal DNA enables inference of the rate at which plasmids accrue mutations (known as the 'rate of evolution') with high precision and accuracy, despite limited genomic information.

## Results

In CoalPT we describe the movement of plasmids between bacterial lineages as a joint coalescent and plasmid transfer process (Fig 1), where lineages can coalesce from present to past or undergo a plasmid transfer event, similar to how recombination is often modeled [25]. The model has two key parameters: the effective population size ($N_e$) and the plasmid transfer rate ($\rho$). These two parameters determine the rates at which coalescent and plasmid transfer events occur. The plasmid transfer events are agnostic about the precise biological mechanism under which plasmids move between bacteria.

The result of the CoalPT model is a timed phylogenetic network with each lineage of the network corresponding to one or more lineages of either the chromosome or plasmid trees. As such, the co-evolutionary history of the chromosome and the plasmid is denoted the timed phylogenetic network in which the chromosome and plasmid trees are embedded. To perform inference under the CoalPT model, we use a Markov chain Monte Carlo (MCMC) sampling technique for the timed phylogenetic network that is related to the MCMC inference of reassortment [26] and recombination networks [27]. Using an MCMC approach allows us to infer the phylogenetic network, effective population sizes, plasmid transfer rates, and evolutionary parameters, all while accounting for uncertainty in the network and parameter estimates.

### Plasmid transfer rates differ depending on plasmid size and function

To investigate how plasmids move between *S. sonnei* lineages, we reconstructed the joint evolutionary history of *S. sonnei* chromosomal DNA and four plasmids (pINV, spA, spB & spC) (Figs 2 and S1). We further considered two additional alignments of the spA plasmid to explore the potential use of this new method in reconstructing the movement of specific mobile AMR elements within populations. This small plasmid contains up to four AMR genes that include *strA* & *B, sul2*, and sometimes *tet(A)* that were established to move between lineages. For spA, the number of reconstructed transfer events strongly depends on which part of the spA plasmid is used for the analyses (S2 Fig). For the first analyses (*entire spA*), we used the entire spA plasmid for isolates that had ≥70% coverage of the spA plasmid, regardless of AMR profile,

PLOS Pathogens

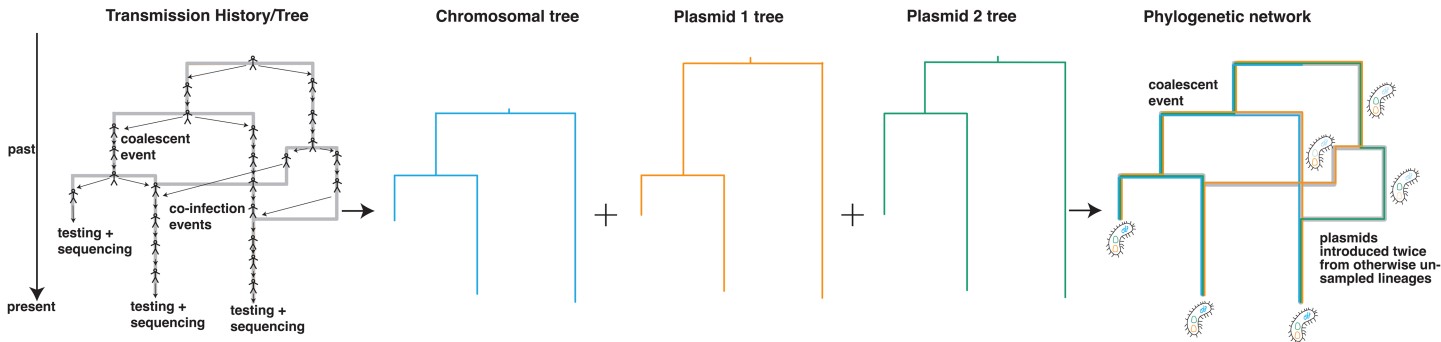

**Fig 1**. **Schematic representation of the coalescent with plasmid transfer model.** The transmission history of bacterial lineages, including the movement of plasmids between lineages, can be described as a transmission network. From that transmission history, we can track the history of the chromosomal and plasmid DNA individually. Here, we track the history of two plasmids and one chromosome from the samples back in time. The history of both chromosomal and plasmid DNA can be described by using a tree. The coalescent with plasmid transfer models a backward-in-time process where any two network lineages can coalesce (share a common ancestor). Additionally, network lineages can undergo a plasmid transfer event, modeled backward in time as one of the plasmid lineages branching off the main branch. How rapidly two lineages share a common ancestor backward in time is given by the effective population size, and the rates of plasmid transfer denote the rate of observing plasmid transfer events backward in time.

with a focus on tracking the plasmid. In a second set of analyses, we used an alignment of the four AMR genes *strA* & *B, sul2* and *tet(A)*, that included the region between *strB* and *tetA*, which also encoded a transposase and *tetR* (*AMR genes only* dataset) (see Methods). Finally, we considered only the genes coding for *strA* & *B* and *sul2* and the flanking region of ~100 bases of *strB* (*strA* & *B* + *sul* + flanking dataset) as these three genes are known to frequently move together. S3 Fig shows the location of these genes on the spA plasmid, and it highlights the regions of the spA plasmid used. We also performed long-read sequencing using ONT platforms of selected isolates (see Methods) that are members of the clade highlighted in red in S4 Fig to determine the location of AMR genes in genomes with different coverage of the spA reference genome (S3 Fig). We made this distinction in part due to evidence that the AMR genes carried on the spA plasmid had integrated into the chromosome for some isolates for which we had lower coverage of the spA plasmid. This lower coverage suggested that these AMR genes were not being carried on the spA plasmid in these genomes (S3 Fig).

To illustrate the differences, we reconstructed a tanglegram from chromosomal and spA timed plasmid trees inferred using IQ-TREE 2 and TreeTime [28–30]. As shown in S4 Fig, there is evidence for multiple rearrangements between the chromosome and plasmid trees, indicating plasmids are moving. The 70% coverage cutoff used for the *entire spA* tree removes a group of isolates highlighted in red in S4 Fig, suggesting either that the genes present on spA changed or that the AMR genes integrated into a different plasmid or the chromosome (S3 Fig).

We next tested whether there is evidence for the different parts of spA to code for different evolutionary histories, or whether they mostly differ by their amount of genomic information. To do so, we analysed isolates that reached the coverage thresholds for all three parts of the spA plasmid as described above. We then treated them each as a separate segment in a phylogenetic network analysis and reconstructed the joint reticulation network describing the shared evolutionary history of the three parts of spA, not finding evidence for the different parts of spA to code for different evolutionary histories (see S5 Fig)

We then reconstructed the movement of the plasmids between bacterial lineages using CoalPT in three different analyses using the different spA alignments. From the overall *S. sonnei* dataset, we subsampled 400 isolates. For the subsampling, we used the number of available plasmid sequences per isolate as the weights in the subsampling, but did not perform any additional subsampling to have even sample numbers over time. This strategy is enriching for isolates that have more plasmids present.

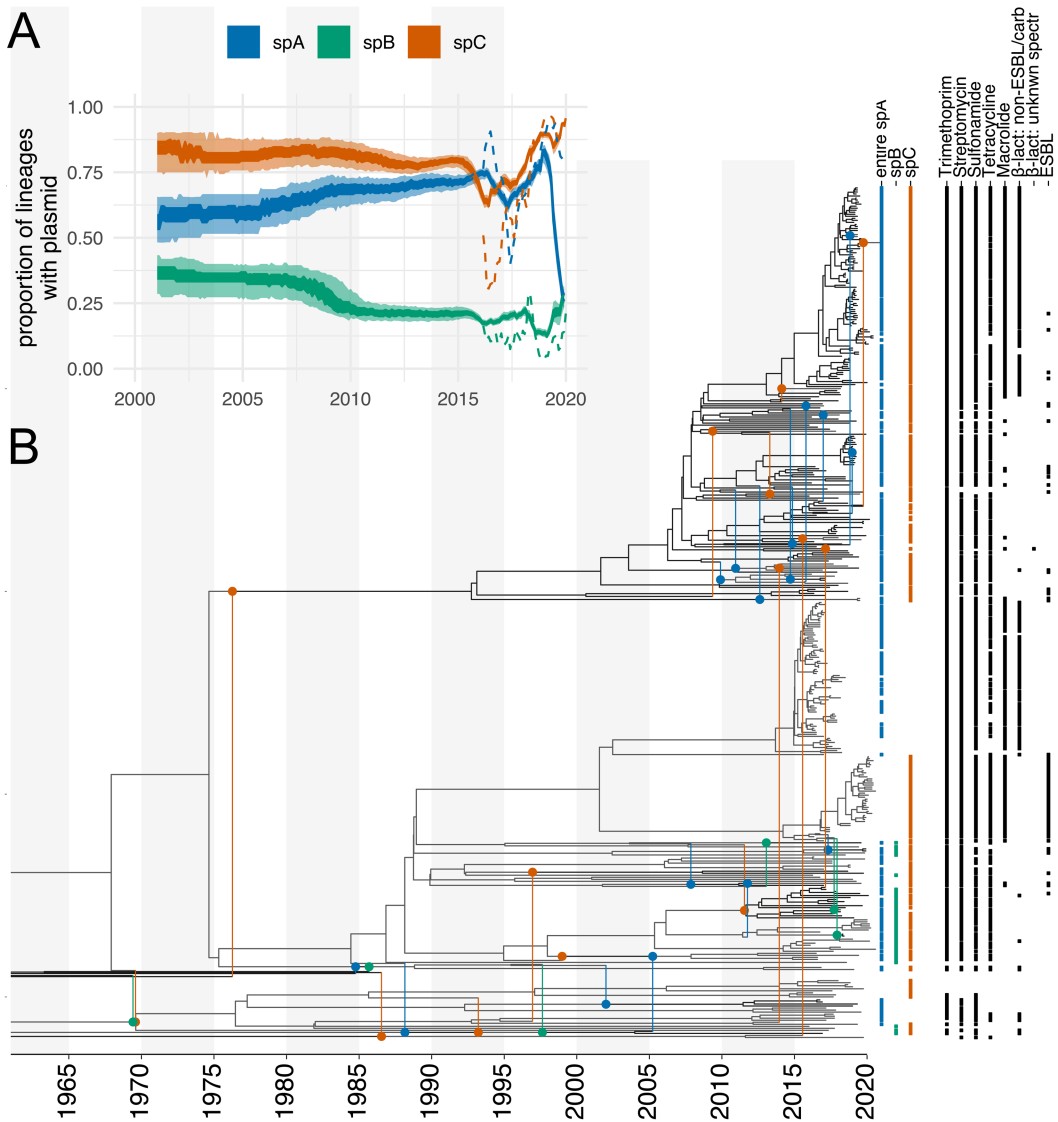

**Fig 2**. **Co-divergence of the core chromosome and plasmids in *Shigella sonnei*. A** Proportion of lineages carrying a plasmid between 2000 and 2020. The inner shaded areas denote the 50% HPD, and the outer area is the 95% highest posterior density (HPD). The dotted lines denote the proportion of samples with a plasmid. **B** Here, we show the maximum clade credibility (MCC) network of *Shigella sonnei* inferred using the chromosomal DNA, the virulence plasmid pINV, and the small plasmids spA, spB, and spC. The MCC network is the network in the posterior distribution that maximizes the sum of clade credibility of the chromosome and plasmid trees. Vertical lines are used to denote plasmid transfer events, where the circles denote the branch to which a plasmid was transferred. The color of the circle denotes either spA, spB, or spC, having jumped between bacterial lineages. The dashed lines correspond to branches from which plasmids branch off. The tip labels in blue, green, and red denote whether a plasmid was detected in a leaf. The black dots denote the presence of antimicrobial resistance determinants to the different drug classes.

Since CoalPT assumes that there is no inter-lineage recombination within chromosomes or plasmids, we masked sections with evidence of recombination in the chromosome but assumed that there is no intralineage recombination on the plasmid alignments (see Methods). We used a separate strict molecular clock and HKY+$\Gamma_4$ [31] substitution model for the chromosome and for each plasmid. We additionally assume a constant-size coalescent process and infer the effective population size and the rate of plasmid transfer, allowing each plasmid to have a different transfer rate.

Modelling changing effective population sizes over time may put a change in the relative weight on longer or shorter networks, which can then lead to slightly different plasmid transfer rate estimates. However, this bias is expected to be small for cases with well-informed rates of evolution, such as provided by the chromosomal sequences here [32], and the rates across the different plasmids will be affected to a similar degree. Further, any bias is likely to be in the same direction for all plasmids. The prevalence of the smaller plasmids was relatively constant over the sampling period (S1C Fig), with spA and spC being frequently detected in different lineages and genotypes of *S. sonnei* (S3 and S6 Figs). In contrast, the spB plasmid was predominantly found in *S. sonnei* isolates belonging to genotypes that are part of global lineage III, although it was detected in six isolates in lineage 2 (S1 Table). We found little to no support for the virulence plasmid, pINV, having been transferred between different bacterial lineages, suggesting co-divergence of the chromosome and pINV (S7 Fig). This finding is consistent with the known evolution of *Shigella* species.

Importantly, the spA plasmid was inferred to have the highest transfer rate between bacterial lineages of *S. sonnei*. The spA plasmid is the only small plasmid that carries AMR determinants (S8 Fig). The other two small plasmids, spB and spC, displayed lower rates of plasmid transfer (S8 Fig). These two plasmids were inferred to have high copy numbers based on relative read depth to the chromosome from Oxford Nanopore Technologies (ONT) data and were not characterized to have any direct ability to mobilize. The number of inferred plasmid transfer events was the highest when using entire spA plasmids and the lowest when only using AMR genes. The same patterns hold when looking at the rate at which plasmids are transferred (S8 and S9 Figs), with the entire spA showing the highest transfer rate.

We next computed the rate at which plasmids are being lost in *S. sonnei*. We first calculated the number of times a plasmid has been lost as the number of child edges (i.e., branches) in a network for which the parent branch carries a plasmid while the child branch itself does not. We then divide this number by the total length of the plasmid tree to get an estimate of the rate at which the plasmid is lost in units of plasmid loss events per unit time. We restrict this analysis to events in the last 10 years since sample collection. We calculate this heuristic for the posterior distribution of networks to estimate the uncertainty of the heuristic. Importantly, the rate at which plasmids are lost is a heuristic based on the reconstruction networks and not directly a model parameter. In the future, this rate could be explicitly incorporated as a model parameter into CoalPT.

The virulence plasmid, which in *S. sonnei* is known to be often lost in culture [14,33], but it forms part of the core genome of all *Shigella* species, had the highest rate of being lost (S9 Fig). This was expected, given the known loss in culture. The smaller plasmids were all lost at a lower rate relative to the pINV plasmid of *S. sonnei* (S10 Fig). These patterns may, at least in part, be driven by the plasmids not being detected or being lost in culture, but may also reflect that these smaller plasmids have limited fitness costs and, as such, are more readily maintained. To further test this point, we performed an analysis where we modeled the presence and absence of pINV, spA, spB, and spC as a continuous-time Markov chain evolving on the chromosome [34]. We compare the rate at which plasmids are gained and lost between the true, observed data, and when we randomly permutate the tip to presence-absence mapping. We find that the rate of gain/loss for pINV is equal between the true and the permuted presence absence (see S11 Fig), indicating that the plasmid is randomly lost during the sampling process, while the three small plasmids show strong differences between the true presence absence and the permuted assignments. We also estimated the individual rates of gain and loss using this approach (see S12 Fig). While this does not fully model the ancestral history of each plasmid, we find that the estimates are broadly consistent with both our estimates using the phylogenetic network inference.

## Accounting for the co-divergence of chromosomes and plasmids is essential to estimating rates of evolution in plasmids

The evolution of the genome of bacterial species occurs, in part, as a result of selective pressures on the core and accessory genomes. The core will likely be under strong selective constraints, while the accessory may be subject to weaker selection. Indeed, we find that plasmids tend to have higher molecular evolutionary clock rates than those of the chromosome (Figs 3 and S13). SNPs within the bacterial chromosome have been the focus of bacterial phylodynamics to date

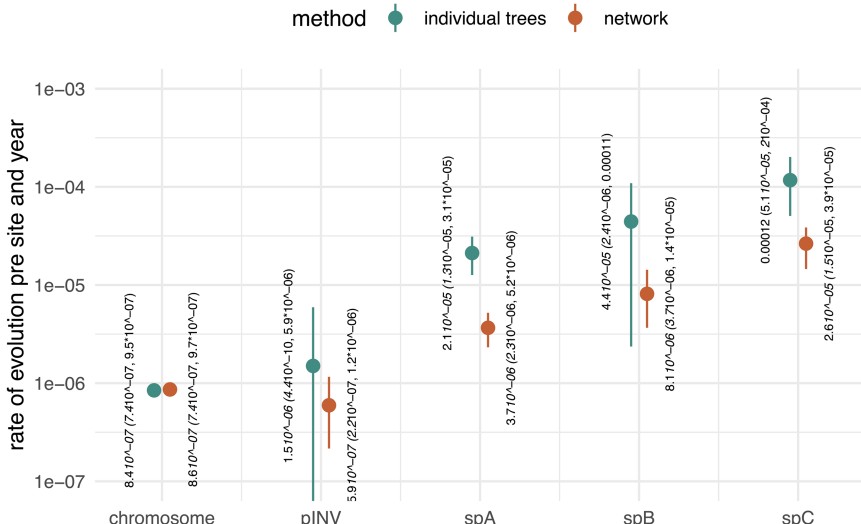

**Fig 3**. **Rates of evolution for plasmids and core chromosome in *Shigella sonnei.*** Here, we compare estimates inferred by assuming an individual rate of evolution in units of substitutions per site per year for the chromosome and plasmids (in green) to those where we explicitly model the joint evolutionary history of these lineages as a phylogenetic network (in orange). The posterior estimates of the evolutionary rate of the chromosomal and plasmid DNA of *S. sonnei* sequences isolated in Melbourne, Australia, over several years. The numbers denote the mean posterior, and values in brackets are the 95% highest posterior densities (HPD).

due to enough temporal signal in the sequence data to model the population dynamics, facilitated by the bacterial chromosome being orders of magnitude larger than some plasmids. In the case of *S. sonnei*, the chromosome has approximately 22 times more nucleotides than the virulence plasmid, pINV, and between ∼570 to ∼2300 times more than spA–spC. In spite of the higher rates of evolution in plasmids, compared to the chromosome, we would still expect an alignment of SNPs in the core chromosome to have a larger number of SNPs due to its sheer size. As such, plasmids are less likely to contain as much information as the chromosome and thus are less likely to behave as measurably evolving populations [35,36]. That is the measurable accumulation of genomic change over the time window of sampling. However, modelling the joint evolution of chromosomes, which are measurably evolving, and plasmids changes this. The timing of past coalescent events where chromosomal and plasmid co-diverged is potentially well informed by the chromosomal DNA, which is measurably evolving. These 'co-divergence' events of chromosomes and plasmids can then act as calibration points to infer the rates of evolution of plasmids.

To illustrate how modeling the co-divergence of the chromosomal and plasmid DNA impacts inferences of the evolutionary rate, we reconstructed the phylogenetic trees of the chromosomes, virulence (pINV), spA, spB, and spC plasmids individually (Fig 3). For the chromosome and the pINV, we used SNP alignments, which only contain the SNPs, in order to reduce the size of the dataset. For spA–spC, we used the full alignments (with gaps, Ns, and both variant and invariant sites) obtained from alignment to the reference genomes (see Methods). We used the same priors and evolutionary models (strict clock model and HKY+$\Gamma_4$) as for the network inference described above and a constant coalescent population model. We then inferred the phylogenetic trees, evolutionary rates, and other parameters. For the network inference, we inferred a separate evolutionary rate for the chromosome and each plasmid. To calculate from the SNP alignment to the whole genome rate of evolution of the chromosome and the virulence plasmid, we multiply the rate estimate based on the SNP alignment by the ratio of SNP sites (11kb) over the whole genome length (4.8Mb).

As shown in S13A Fig, we found the chromosome to evolve at a rate with mean $7.8 \times 10^{-7}$ subs/site/year (95% highest posterior density, HPD $6.6 \times 10^{-7} - 9.4 \times 10^{-7}$), and the virulence plasmid to evolve at a rate with mean $9.3 \times 10^{-7}$

subs/site/year (95% HPD $5.8 \times 10^{-7} - 1.3 \times 10^{-6}$). The small plasmids spA–spC all evolve at substantially higher rates, with means of between $2.9 \times 10^{-6}$ and $1.9 \times 10^{-5}$ subs/site/year. Importantly, inferring these rates of evolution would be impossible using the plasmid alignments alone and thus requires information about the co-divergence of the plasmids and the chromosome.

To further explore the impact of our approach on estimates of evolutionary rates, we compared the inferred rates for plasmids using the coalescent with plasmid transfer and individual tree inference using simulations. As shown in S14 Fig, using tree inference only to retrieve rates of evolution will return the prior on the evolutionary rate, even for cases with relatively many SNPs, implying that the data are not sufficiently informative to drive the estimate of this parameter. The reason is that even in cases with many SNPs in total, the number of SNPs per time that one expects to occur over the sampling period of five years is only $5\,\text{years} \times 200\,\text{bp} \times 5 \times 10^{-4}$ subs/site/year = 0.5 SNPs for the largest plasmid. The network approach, on the other hand, is able to infer the rates of evolution of plasmids even when only a few SNPs occur (S13B Fig) because the chromosome data act as a form of molecular clock calibration and thus there is more data available for inference. Lastly, we tested whether the strict clock assumption we used for the chromosomal data was valid. We find some evidence for a higher coefficient of variation describing rate heterogeneity across lineages in the true data compared to simulated data under a strict clock, but the confidence intervals overlap (see S15 Fig). This indicates that there may be a slight variation in evolutionary rates among lineages that is not fully accounted for using the strict clock model.

## Evidence for cross-species MDR plasmid exchange and steady growth of pKSR100 prevalence

We next investigated the movement of an MDR plasmid that has been previously well-characterized using genomic epidemiological approaches to be moving within and between lineages of two *Shigella* species, *S. sonnei* and *S. flexneri*. To do so, we compiled three alignments. We made an alignment from SNPs in the reference chromosome for both *S. sonnei* (n = 789 isolates) and *S. flexneri* (n = 297 isolates) individually (see Methods). For the MDR plasmid (pKSR100) known to circulate in both species, we aligned sequences from both species jointly. All *S. sonnei* and *S. flexneri* isolates that had ≥80% coverage of the reference plasmid were included in the alignment and a core SNP alignment with ≥95% conservation of sites (S16 Fig) (see Methods for plasmid threshold selection for alignments). We then randomly sub-sampled 250 isolates equally from *S. sonnei* and *S. flexneri* that carried the pKSR100-like plasmids (250 isolates were selected due to computational limitations). The 250 samples were chosen based on computational limitations. The chromosomal DNA of *S. sonnei* and *S. flexneri* were assumed to be their individual trees, while all samples of the pKSR100 plasmids were assumed to be from the same trees. We next reconstructed the joint evolutionary history of the core chromosome and the MDR plasmid, assuming a strict molecular clock for both the chromosome and the MDR plasmid and an HKY+$\Gamma_4$ substitution model. In order to improve the computational efficiency, we fixed the rate of evolution of the core chromosomes to be equal to the estimates in S13 Fig, while estimating the rate of evolution of the MDR plasmid.

We found evidence for multiple events where the MDR plasmid jumped between bacterial lineages within species and also between species (Fig 4 and S17 Fig). These jumps between lineages were, in some cases, associated with a rapid expansion of a clade. For example, we found that the *S. sonnei* clade expanded after the introduction of an MDR plasmid into the bacterial lineage from *S. flexneri* around 2010. We next sought to distinguish introductions of the MDR plasmid into *S. sonnei* and *S. flexneri* clades by whether they likely originated from the other bacterial species or from an unknown species entirely. To do so, we followed the procedure described in *Directionality of plasmid transfer*. Additionally, we only considered plasmid transfer events that were introduced into *S. sonnei* or *S. flexneri* in the last 50 years. There is evidence for multiple introductions of plasmids into both species from each other (Fig 4), but also from unknown bacterial lineages, which could be other *Shigella* lineages or from other bacterial species in the same ecological niches, as has been

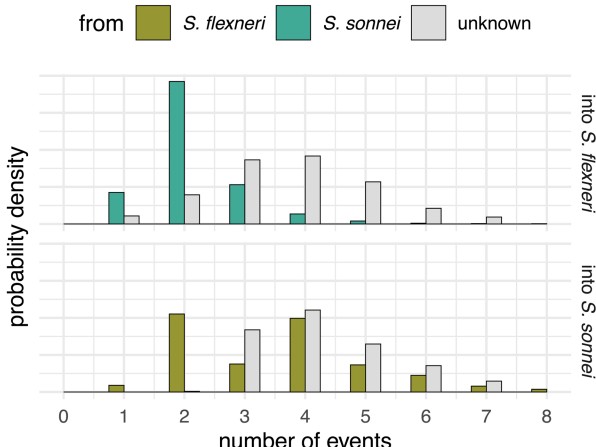

**Fig 4. Estimated number of cross-species movements of pKSR100 between *S. sonnei* and *flexneri*.** Here, we show the posterior estimates for the number of times where a plasmid moved between *S. sonnei* and *S. flexneri*. The assignment of directionality is based on the chromosome. For example, if the parental node at a plasmid transfer event has a chromosome belonging to *S. sonnei*, the origin is considered *S. sonnei*.

previously reported [37]. We next computed the proportion of lineages in the past that carried the plasmid pKSR100. As shown in Fig 5, we find a steady increase in the proportion of bacterial lineages that carry the pKSR100 plasmid. This increase is inferred to start around the year 2010 and to continue relatively steadily until 2020, when we have the most recent samples in the dataset.

## Discussion

Our novel inference approach represents a substantial advancement in the field of bacterial population genomics by enabling a more comprehensive understanding of the plasmid movements within bacterial pathogens over time from routinely generated WGS short-read data. To date, studies exploring the movement of specific plasmids have relied on long-read sequencing data and have been undertaken in localized settings over shorter timespans [4,10,38]. CoalPT represents an approach that can leverage the short-read data generated from genomic surveillance efforts in public health laboratories, providing a method to explore the movement and transfer of plasmids in bacterial datasets, all while accurately accounting for uncertainty in the data.

Reconstructing the movement of plasmids over time has been difficult, but it is increasingly of interest to better understand the evolutionary dynamics shaping potential plasmid-driven outbreaks. CoalPT is designed for tracking plasmids, with a complete reference sequence, meaning that the quality of plasmid alignments warrants special attention. Our approach of varying the coverage of the plasmid, core-SNP filter, and use of the consistency index for quantifying homoplasy [39] can help guide assessments of the impact of data quality on inference of plasmid dynamics. In line with other research, we find a high degree of co-divergence of virulence plasmid, pINV, with the chromosome of *S. sonnei* [14], and the movement of small plasmids within the *S. sonnei* population. Furthermore, we find evidence for multiple MDR plasmid transfer events between *S. sonnei* lineages, but also between *S. sonnei* and *S. flexneri* lineages [12,13,19,22]. The model behind CoalPT offers the potential for integrating geographical data through phylogeographic approaches [34,40]. Such advances will be critical in exploring, monitoring, and potentially modeling future scenarios of how drug-resistant plasmids of interest are moving in large-scale datasets of high-priority pathogens. Explicitly modeling the co-divergence of plasmids and core genomes also allows us to quantify the number of these events, the timing of introductions, and the

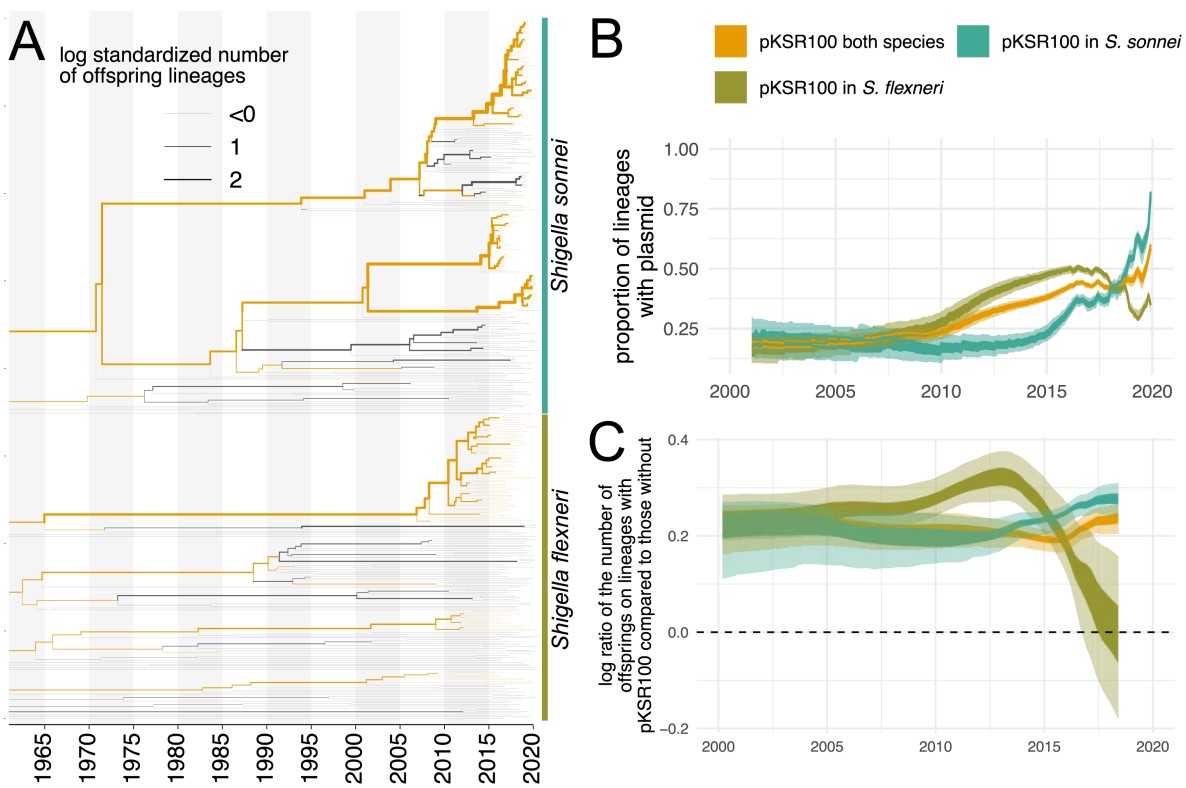

**Fig 5. Frequency of *S. sonnei* and *S. flexneri* clusters with and without pKSR100 plasmid. A** Chromosomal trees of *S. sonnei* and *S. flexneri* with the presence or absence of pKSR100 mapped onto the plasmid tree. The width of the branches denotes the log standardized cluster size below a node relative to all other co-existing lineages. **B** Proportion of lineages with and without pKSR100 since 2020. The inner interval denotes the 50% highest posterior density (HPD) and the outer the 95% computed using the logged trees. **C** The log standardized cluster sizes below nodes with pKSR100 compared to co-existing lineages without pKSR100.

lineages from which plasmids originate that are later introduced into other lineages or species. As such, this framework is amenable to studying other bacterial populations where the plasmid dynamics are less clear.

We note that our method to quantify plasmid transfer rates does have some limitations. It is suited to the exploration of specific plasmids of interest, and not suited to tracking unknown plasmids. These approaches would be immediately relevant to drug-resistant plasmids that may be driving population expansions, but could also be extended to virulence plasmids or where there has been reported convergence of AMR and virulence [41]. We demonstrated the utility of this novel method for the reconstruction of the movement of plasmids, but also showed the limitation of this approach, particularly for small AMR mobile elements, where there may be a lack of phylogenetic signal. As such, it is not suited for tracking the movement of specific AMR genes within populations in the absence of any other phylogenetic context. Instead, our approach addresses the question of whether the expansion of a particular plasmid was due to a single introduction and subsequent expansion, or due to repeated introductions from different sources. Finally, there are uncertainties around estimating the length of time some plasmids may have spent with their hosts. However, this approach provides the basis that these methods may be further refined to better capture the impacts of different plasmids in shaping bacterial populations. Such insight is key to better understanding the complex evolutionary dynamics of plasmids and their role in the emergence of drug resistance and virulence. There is extensive evidence of plasmid movement between bacteria of distant lineages or species (e.g. [42]). Currently, the movement of plasmids between different species can be modelled by combining different chromosomal alignments into a single alignment, as we have done here for *S. sonnei* and *S. flexneri*

to reconstruct the movement of pKSR100 between the two species. However, we had to assume a shared coalescent process. Future work should explore these processes, potentially using a hierarchical network modelling structure similar to the multispecies coalescent [43].

Modeling plasmid evolution has profound implications for calibrating their molecular clock and inferring their evolutionary rates and timescales. The main factors to consider for molecular clock calibration are sequence sampling times and the amount of information that accumulates over time, where the latter pertains to the product of the evolutionary rate and the number of sites. Our results show that plasmid sequence data alone are insufficient to calibrate the molecular clock, such that joint analyses of chromosome and plasmid data, as in CoalPT, are essential for understanding plasmid evolution.

Our current implementation does not model potential differences in fitness between lineages that carry plasmids and those that do not [44]. This effect could, in principle, be modeled to study the fitness benefits and costs of plasmids on a population level by treating the fitness of a lineage as a function of the presence or absence of a plasmid [45]. Such analyses would be particularly interesting in the context of empirically measured fitness costs in culture. An additional insight that could be gained is how plasmids are introduced and transferred between different host types, by extending the current unstructured coalescent approach to account for population structure [40,46,47].

Finally, we showed that modeling the co-divergence of plasmid and chromosomal DNA allows us to reconstruct the plasmid phylogeny much more precisely. In turn, these inferences improve the accuracy with which we can unravel key evolutionary pathways, such as the timing of their introduction to a population and the timescale of point mutations of epidemiological relevance. Importantly, the only source of evolutionary information that we consider is point mutations. Novel approaches that integrate, for example, structural rearrangements of plasmids, could provide additional insight into the evolutionary dynamics of plasmids moving within and between lineages. Such approaches would have applications to better understand the movement of drug-resistant plasmids both locally, within specific clinical settings, or internationally, such as tracking the dissemination of plasmids of interest across the globe.

## Methods

### Ethics statement

Data were collected in accordance with the Victorian Public Health and Wellbeing Act 2008. Ethical approval was received from the University of Melbourne Human Research Ethics Committee (study number 1954615.3 and reference number 2024-30320-59894-3).

### Coalescent with plasmid transfer

Bacterial lineages can exchange plasmids through different mechanisms. To model this process, we use a coalescent-based model related to the coalescent with re-assortment [26]. In the coalescent with plasmid transfer model, we model a backward-in-time process starting from sampled individuals (Fig 1). We used a separate strict molecular clock and HKY+$\Gamma_4$ [31] substitution model for the chromosome and for each plasmid. The sampled individuals are required to have a chromosome but can have anywhere from 0 to $n$ plasmid sequences. For a given effective population size, $N_e$, and plasmid transfer rate $\rho$, we sample the time to the next coalescent event (from present to past) from an exponential distribution with a rate of $\binom{|L|}{2}\frac{1}{N_e}$, where $|L|$ denotes the current number of lineages. The time until the next plasmid transfer event, for each individual plasmid, is drawn from an exponential distribution with mean $\frac{1}{|L|*\rho}$, with $\rho$ being the plasmid-specific transfer rate and $|L|$ being the number of lineages that have both the chromosome and the plasmid, or more than one plasmid, if the chromosome is absent. Upon a coalescent event, the parental lineage will carry the union of chromosomal and plasmid lineages of the two child lineages. Upon a plasmid transfer event, one plasmid lineage is randomly chosen to branch off into one parental lineage, whereas all other plasmids and the chromosomal lineages will follow the

other parental lineage. This is different from how re-assortment is modeled in [26] in that a plasmid transfer occurs relative to the chromosome, and only one plasmid is transferred at a time. In other words, at each plasmid transfer event, we say one plasmid has originated from a different parental lineage than the chromosome. The backwards-in-time rate of a plasmid having originated from a different parental lineage models the forward-in-time process where exactly one plasmid moves from one bacterium to another. The method is agnostic to how a plasmid is transferred, other than the assumption that only one plasmid is transferred at a time. However, we assume that there is no interlineage recombination within the chromosomal or plasmid DNA, although this is an assumption that could potentially be relaxed in the future by employing a similar approach to [27]. Importantly, the resulting phylogenetic network is not constrained to be tree-based (as e.g [48,49]) but allowed to have any possible structure one can simulate under the coalescent with plasmid transfer.

Under a coalescent process, sampling intensity is generally assumed to be low, relative to the effective population size [50]. This assumption applies to the sampling at the level of the isolates in our model. Each isolate is assumed to have at least the chromosomal sequence. In cases where isolates do not have the full set of plasmids, the model currently treats the plasmids as missing data rather than explicitly modelling the loss of plasmids. However, an important distinction is that even when the plasmid transfer and coalescent parameters are correctly estimated, the number of plasmid transfer events inferred refers to those that explain the samples collected, and not necessarily those of the whole population, a situation that is analogous to discrete phylogeographic models [51].

## Posterior distribution

In order to perform joint Bayesian inference of phylogenetic networks, the embedding of chromosome and plasmid trees, together with the parameters of the associated models, we use an MCMC algorithm to characterize the joint posterior density. The posterior density is denoted as:

$$P(\mathcal{N}, \mu, N_e, \rho | D) = \frac{P(D|\mathcal{N}, \mu)P(\mathcal{N}|N_e, \rho)P(\mu, N_e, \rho)}{P(D)}, \tag{1}$$

where $\mathcal{N}$ denotes the network, $\mu$ the parameters of the molecular substitution and clock model, $N_e$ the effective population size and $\rho$ the plasmid transfer rate. The coalescent model $N_e$ can be any model that describes an effective population size over time, meaning it can describe a constant rate coalescent process (constant $N_e$) or parametric or non-parametric $N_e$ dynamics. The plasmid transfer rate is currently assumed to be constant over time, but can vary between different plasmids. The multiple sequence alignment, that is, the data, is denoted $D$. $P(D|\mathcal{N}, \mu)$ denotes the network likelihood, $P(\mathcal{N}|N_e, \rho)$, the network prior and $P(\mu, N_e, \rho)$ the parameter priors. As is usually done in Bayesian phylogenetics, we assume that $P(\mu, N_e, \rho) = P(\mu)P(N_e)P(\rho)$.

As is the case in coalescent approaches on trees, we condition on the sampling event, meaning that we assume the sampling events occur at predefined times. As such, the assumptions of the coalescent process apply, such as that a very small fraction of the overall population is sampled. The assumption applies at the isolate level. On the level of plasmids and chromosomes, we assume that for each type of plasmid, we have one sample per isolate. For isolates for which we do not have a specific version of a plasmid, we treat that plasmid as missing information, rather than explicitly modelling its absence. The coalescent events are assumed to happen independently of plasmid transfer events. In particular, this assumes there are no couple of coalescent/plasmid transfer events, which become more likely for very small population sizes [52].

**Network likelihood.** As we assume that there is no between-lineage recombination within the chromosomal or plasmid DNA, we can simplify the network likelihood $P(D|\mathcal{N}, \mu)$ into the tree likelihood of the chromosomal and plasmid DNA. If $T_i$ is the tree of the chromosome or plasmid (with $i = 0$ being the chromosome tree and $i>0$ being plasmid trees) and if $D_i$

is either the chromosomal or plasmid alignment, we can write the network likelihood as:

$$P(D|\mathcal{N}, \mu) = \prod_{i=0}^{chromosome+nr\ plasmid} P(D_i|T_i, \mu), \qquad (2)$$

The tree likelihood calculations use the default implementation of the tree likelihood in BEAST2 [24] and can use beagle [53] to increase the speed of likelihood calculations. Importantly, this approach allows us to use all the default substitution and clock models in BEAST2, including, for example, relaxed clock models discussed here [24].

**Network prior.** The network prior is denoted by $P(\mathcal{N}|N_e, \rho)$, which is the probability of observing a network and the embedding of chromosomal and plasmid trees under the coalescent with plasmid transfer model. $N_e$ denotes the effective population size, and $\rho$ the per plasmid transfer rate. The network prior is the equivalent of the tree prior in phylogenetic tree analyses.

We can calculate $P(\mathcal{N}|N_e\rho)$ by expressing it as the product of exponential waiting times between events (i.e., plasmid transfer, coalescent, and sampling events):

$$P(\mathcal{N}|N_e, \rho) = \prod_{i=1}^{\#events} P(event_i|L_i, N_e, \rho) \times P(interval_i|L_i, N_e, \rho), \qquad (3)$$

where we define $t_i$ to be the time of the i-th event and $L_i$ to be the coexisting lineages extant immediately prior to this event.

Given that the coalescent process is a constant-size coalescent and given the i-th event is a coalescent event, the event contribution is denoted as:

$$P(event_i|L_i, N_e, \rho) = \frac{1}{N_e}. \qquad (4)$$

If the i-th event is a plasmid transfer event and assuming a constant rate over time, the event contribution is denoted as:

$$P(event_i|L_i, N_e, \rho) = \rho. \qquad (5)$$

This event contribution can be generalized to account for different rates of transfer for different plasmids by substituting $\rho$ with the plasmid-specific rate depending on which plasmid was transferred. The interval contribution denotes the probability of not observing any event in a given interval. It can be computed as the product of not observing any coalescence nor any plasmid transfer event in the interval $i$. We can, therefore, write:

$$P(interval_i|L_i, N_e, \rho) = exp[-(\lambda^c + \lambda^r)(t_i - t_{i-1})], \qquad (6)$$

where $\lambda^c$ denotes the rate of coalescence and can be expressed as:

$$\lambda^c = \binom{|L_i|}{2}\frac{1}{N_e}, \qquad (7)$$

and $\lambda^r$ denotes the rate of observing a plasmid transfer event on any co-existing lineage and can be expressed as:

$$\lambda^r = \rho \sum_{l \in L_i} \mathcal{L}(l) * \begin{cases} n_i, & \text{if lineage has chromosome or } n_i > 1 \\ 0, & \text{else if } n_i = 1 \end{cases} \qquad (8)$$

with $n_i$ being the number of plasmids on $\mathcal{L}_i$ and $L_i$ existing in that interval. What the above equation means is that we can observe a plasmid transfer event only on lineages that have either the chromosome or more than one plasmid present, as we can only observe the movement of plasmids relative to other plasmids or the chromosome.

**MCMC algorithm for plasmid transfer networks.** In order to infer the network topology, timings of individual events, as well as embedding of chromosome and plasmid trees within the plasmid transfer network, we employ Markov chain Monte Carlo sampling of the networks and embedding of trees. This MCMC sampling employs operators that operate on the network topology, embedding of trees within that network, or the timings of individual events, such as coalescent or plasmid transfer events. The operators we use are similar to the ones used in [26] and in [27], but the condition is that only one plasmid jumps between bacterial lineages at a time.

## Operator descriptions

**Add/remove operator.** The add/remove operator adds and removes plasmid transfer events. The add/remove operator on networks is an extension of the subtree prune and regraft move for networks [54]. Similar to [27], we also added an adapted version to sample re-attachment under a coalescent distribution to increase acceptance probabilities. We show an example of such an add/remove step in S19 Fig.

**Exchange operator.** The exchange operator changes the attachment of edges in the network while keeping the network length constant.

**Subnetwork slide operator.** The subnetwork slide operator changes the height of nodes, that is, the time from a node relative to the most recent sampled individual, in the network while allowing the change in the topology.

**Scale operator.** The scale operator scales the heights of the root node or the whole network without changing the network topology.

**Gibbs operator.** The Gibbs operator samples any part of the network that is older than the root of any segment of the alignment and is thus not informed by any genetic data and is the analog to the Gibbs operator in [26] for re-assortment networks. As a Gibbs operator, this move is always accepted

**Empty edge preoperator.** The empty edge preoperator augments the network with edges that do not carry any loci for the duration of a move, to allow for larger jumps in network space.

The roots of phylogenetic networks can be much more distant than the roots of the individual plasmid trees. As in [27], we assume the plasmid transfer rate to be reduced prior to the individual plasmid trees having reached their root. As shown in [27], this assumption does not affect parameter inference.

For parameter inference, such as the inference of the $N_e$ and $\rho$ or the evolutionary parameters, we use the operators already implemented in BEAST2

## *Shigella* dataset

*S. sonnei* (n = 789) and *S. flexneri* (n = 297) isolates (excluding *S. flexneri* serotype 6 isolates) received at the Microbiological Diagnostic Unit Public Health Laboratory (MDU PHL, the bacteriology reference laboratory for the state of Victoria, Australia), between January 2016 and December 2020, were included in this study. These isolates were accompanied by the year and month of collection. These isolates undergo routine WGS on Illumina NextSeq platforms using DNA extraction and sequencing protocols previously described [18].

**Reference genomes.** The complete reference sequences for *S. sonnei* strain Ss046 and *S. flexneri* 2a strain 301 were downloaded from NCBI. For *S. sonnei* strain Ss046, these included the chromosome (accession number NC 007384), the virulence plasmid, pINV (accession number NC 007385 214,396 bases), the spA plasmid (accession number NC 009345 8,401 bases), the spB plasmid (accession number NC 009346 5,153 bases), and the spC plasmid (accession number NC 009347 2,101 bases). For *S. flexneri* these included the chromosome (accession number NC 004337) and the virulence plasmids pINV (accession number NC 004851 221,618 bases). The complete reference for

the pKSR100 plasmid pKSR100 strain SF7955 (accession number LN624486, 73,047 bases) was also downloaded. This MDR plasmid has been found in *S. sonnei* and *S. flexneri* lineages circulating in MSM populations since 2015 [12].

The complete plasmid genomes were characterized *in silico* for replicon family, relaxase type, mate-pair formation type, and predicted transferability of the plasmid using mob-suite (v3.0.2) [55]. This confirmed the pINV and large MDR plasmid were all IncF plasmids, consistent with previous studies [12,14,22]. The predicted mobility of two of the small *S. sonnei* plasmids was typed as mobilizable (spA and spB), while spC was typed as non-mobilizable. The presence of known AMR genes on the plasmids was confirmed using the ISO-accredited abritAMR with the species flag for *E. coli* [56]. This confirmed *bla*TEM-1, *erm(B)* and *mph(A)* on the pKSR100 plasmid, and *strAB, tet(A)* and *sul2* on spA.

**SNP alignments of the chromosome.** Alignments of the core genome were generated for both the *S. sonnei* and *S. flexneri* isolates. While the virulence plasmid constitutes part of the core genome for *Shigella* species, it was not included in the core SNP analysis as the pINV in *S. sonnei* is frequently lost in culture. The 789 *S. sonnei* were aligned to the reference *S. sonnei* chromosome Ss046 (accession number NC 007384) to call SNPs using Snippy v4.4.5 (https://github.com/tseemann/snippy), with filtering of phage regions identified using PHASTER [57] and recombination detection undertaken with Gubbins (v2.4.1) [58]. SNPsites (v2.5.1) [59] was used to extract the variant SNPs, resulting in a SNP alignment of 7,793. The genotype for the *S. sonnei* isolates was also determined using the Sonneityping scheme [60].

The same approach was used for the 297 *S. flexneri* isolates using the reference *S. flexneri* 2a str 301 (accession number NC 004337), resulting in a SNP alignment of 21,311 sites.

**Generation of plasmid alignments for *S. sonnei* and *S. flexneri*.** All 789 *S. sonnei* were also aligned to the four plasmids of Ss046 using Snippy v4.4.5. To determine the optimal alignments for the different plasmids from the short-read data, three approaches were explored. First, the percentage coverage of the plasmid reference by the *Shigella* isolates was determined, with different alignments generated using snippy-core for isolates that had $\geq$60% coverage, $\geq$70% coverage, $\geq$80% coverage and $\geq$90% coverage of plasmids. Given that snippy-core has a strict core threshold, variation in the core threshold in the different plasmid alignments was explored using Core-SNP-filter (https://github.com/rrwick/Core-SNP-filter) [61]. Alignments were generated using 0.80, 0.85, 0.90, 0.95, and 1.00 core thresholds from the full snippy alignments for the large plasmids (pINV and pKSR100). To determine the optimal balance of plasmid reference coverage and core SNP threshold for plasmid alignment, the Consistency Index implemented in the R package phangorn was used [62]. The Consistency index is defined as the minimum number of changes/number of changes required on the tree. A Consistency Index of 1 indicates that there is no homoplasy. Maximum likelihood trees were inferred for each alignment using IqTree2 (v2.1.4-beta) [63] with a model of GTR+G and 1000 bootstraps. These were used as input, along with the alignment, to determine the Consistency Index. The final alignment for the *S. sonnei* pINV with 57 isolates was $\geq$60% coverage and a core SNP alignment with $\geq$95% conservation, and a Consistency Index $\geq$0.90. The final alignment for the pKSR100 alignment with both *S. sonnei* and *S. flexneri* isolates (n = 560) was $\geq$80% coverage and a core SNP alignment with $\geq$95% conservation, and a Consistency Index $\geq$0.80.

For the three small *S. sonnei* plasmids, different core thresholds and plasmid reference coverage were considered. The plasmid reference coverage threshold for spA was $\geq$70% coverage (entire spA), while a higher threshold of $\geq$90% coverage was used for spB and spC based on the binomial distribution of plasmid coverage. Given the small size of these plasmids and the low number of SNPs, the full alignment with all variant sites was used, effectively a core threshold of 0.00. This included the gaps and Ns in the sites in the alignments. A total of 555 isolates were included in the spA alignment, 94 isolates for the spB alignment, and 572 isolates for the spC alignment. The final alignments had a Consistency Index of $\geq$0.85 for spA and 1 for spB and spC.

To determine the potential use of this approach for tracking AMR genes as opposed to plasmids, two additional alignments of the AMR genes carried on spA were also generated for 452 *S. sonnei*. Isolates were included in both alignments if *strA* & *B* and *sul2* were confidently detected (partial hits were not considered as these were unlikely to be

functional). For the first alignment snippy-core was used for the spA plasmid with masking to include the region spanning from *sul2* to *tetA* (AMR genes). For the second additional alignment, snippy-core was used for the spA plasmid with masking to only include from *sul2* to the end of the plasmid reference (and additional ∼100 bases flanking *strB*) (*strA*, *strB*, *sul2* and flanking) For both of these, the full alignment with all variant, invariant, and masked sites was used.

**Draft genome assembly and characterization of AMR.** All *S. sonnei* and *S. flexneri* isolates were assembled with Unicycler v0.5.0 [64]. The draft genomes were screened for known AMR determinants using abritAMR [56] with the *E. coli* species flag. The genes detected are either 'exact matches' (100% identity and 100% sequence coverage compared to the reference protein sequence) or 'close matches' (90–100% identity and 90–100% sequence coverage compared to the reference protein sequence, marked by an asterisk [*] to distinguish from exact matches). The assembly graphs from Unicycler were initially explored in isolates that had ≥90% coverage of the small plasmids spB and spC and were found to be circularised in the short-read data. This was explored further using Circular-Contig-Extractor (https://github.com/rrwick/Circular-Contig-Extractor) that takes a GFA assembly graph as input and extracts complete circular contigs.

**Long read sequencing, assembly, and exploration of AMR of selected representative isolates.** A total of 31 *Shigella* isolates underwent long-read sequencing on the Oxford Nanopore Technologies (ONT) platforms. The selected isolates for ONT (23 *S. sonnei* and 8 *S. flexneri*) represented the different AMR profiles, genotype (for *S. sonnei*) and tree structure (for *S. flexneri*). The isolates were cultured overnight at 37°C Luria-Bertani (LB) Miller agar. DNA was extracted using GenFind V3 according to the manufacturer's instructions (Beckman Coulter). The SQK-NBD112.96 kit was used for sequencing libraries. Isolates were sequenced on the R10 MinION flow cells with base calling by Guppy v3.2.4. The long read data were assembled using Unicycler v0.5.0 using the hybrid approach [64]. The assembly graphs from Unicycler were investigated for the location of the AMR genes of interest using Bandage [65]. BRICK was used to visualize the plasmids (https://github.com/esteinig/brick).

**Validation and testing.** Phylogenetic networks sampled under the coalescent with plasmid transfer should describe the same distribution as those simulated under the coalescent with plasmid transfer. As such, we compare the distributions of networks simulated under a set of parameters to those sampled using MCMC under the same set of parameters (in other words, to those sampled under the prior). If the implementation of the MCMC is correct, the two distributions of networks should match. As shown in S18 Fig, the sampled and simulated network distributions match.

We next performed a well-calibrated simulated study, where we simulated phylogenetic networks under effective population size and plasmid transfer rates sampled from the prior. We then infer the effective population sizes, plasmid transfer rates, and phylogenetic networks using, as priors, the same distributions used to sample the parameters for simulations. As shown in S20 and S21 Figs, we can retrieve the effective population sizes and plasmid transfer rates from simulated datasets.

**Directionality of plasmid transfer.** In order to estimate the directionality of plasmid transfers, we first classify each network lineage that carries the information of a chromosome into either *S. sonnei* or *S. flexneri*, based on the chromosome. Each reticulation event, which corresponds to a plasmid being introduced into a new bacterial lineage, is then classified based on the chromosome assignment, telling us into which species a plasmid has been introduced. For example, a plasmid being transferred onto a network lineage with the chromosome belonging to *S. sonnei* is classified as an introduction into *S. sonnei*.

We then infer that a plasmid has originated from *S. sonnei* or *S. flexneri* if the plasmid lineage has originated from a chromosomal lineage belonging to either species or from an unknown species entirely. To do so, we follow the plasmid lineage at each reticulation event backwards in time until we reach the next coalescent event of that plasmid lineage with another plasmid lineage. If this coalescent event has a corresponding chromosomal lineage, we say the plasmid originated from the species this lineage belongs to. As we do not explicitly consider plasmids other than *S. sonnei*

or *S. flexneri*, we further assume that a plasmid has originated from an unknown species if this coalescent event is more than 50 years in the past.

### Cluster size comparison

To get a measure of the relative fitness of lineages with and without pKSR100, we compare the number of offspring of lineages with and without the plasmid. To do so, we first map the presence and absence of pKSR100 onto the chromosomal tree. For each lineage, we then count the total number of leaves in the cluster below that node. For each node in the tree, we then compare the number of leaves below that node to the number of leaves below any other co-existing lineage. Next, we log-standardize the cluster sizes across these co-existing lineages and then compare all nodes with and without pKSR100. We do this once for the entire tree and once only for *S. sonnei* and *S. flexneri* lineages. We repeat this for all the logged iterations in the posterior distribution to get the 95 % highest posterior density interval across the different iterations.

### Visualization

All plots were generated in R using ggplot2 [66], colorblindr [67], and ggpattern [68]. Convergence of MCMC chains was assessed using coda [69]. The networks and tanglegrams are plotted using an adapted version of baltic https://github.com/evogytis/baltic/.

### Supporting information

**S1 Table. Details of *Shigella sonnei* dataset.**
(CSV)

**S2 Table. Details of *Shigella flexneri* dataset.**
(CSV)

**S1 Fig. Co-divergence of the core chromosome and plasmids in *S. sonnei* when only using the AMR genes instead of the entire spA plasmid A** Proportion of lineages carrying a plasmid between 2000 and 2020. The inner shaded areas denote the 50% HPD, and the outer area is the 95% HPD. The dotted lines denote the proportion of samples with a plasmid. **B** Here, we show the maximum clade credibility (MCC) network of *Shigella sonnei* inferred using the chromosomal DNA, the virulence plasmid pINV and the small plasmids spA, spB, and spC. Vertical lines are used to denote plasmid transfer events, where the circles denote the branch to which a plasmid was transferred. The color of the circle denotes either spA, spB, or spC, having jumped between bacterial lineages. The dashed lines correspond to branches from which plasmids branch off. The text denotes the posterior probability of plasmid transfer events for events with a posterior support of over 0.5. **C** The tip labels in blue, green, and red denote if a plasmid was detected at a leaf. The black dots denote the presence of antimicrobial resistance to the antimicrobials on top.
(PDF)

**S2 Fig. Comparison of the posterior distribution of plasmid transfer events when using different parts of the spA plasmid.** Here, we compare the posterior distribution of the number of plasmid transfer events when using different parts of the spA plasmid for inference. Each violin plot is created from a different analysis using either the entire spA plasmid, the combination of four AMR genes from *sul2* to *tetA*, the three AMR genes *sul2, strA, strB*, and the flanking region of ~100 bases.
(PDF)

**S3 Fig. Overview of spA detection in the *S. sonnei* data.** Here, we show the approach for the spA alignment. Panel A) shows the percentage cover of the reference plasmid. Panel B) shows the number of SNPs detected at four different

percentages of reference thresholds and different core SNP filter thresholds. Panel C) shows the visualization of contigs from ONT assemblies for representative isolates AUSMDU00029307 ($\geq$90), AUSMDU00020566 ($\geq$70), and AUSMDU00036386 ($\geq$50) at different percentage cover of the reference plasmid. Panel D) the complete genome assembly of AUSMDU00036386, which shows a single chromosome and four plasmids. The blue lines in the plasmid of 107,245 bases show the blast hits for the two AMR regions in spA.
(PDF)

**S4 Fig. Tanglegrams of the chromosome, and reconstructed trees when using different parts of the spA plasmid.** Here, we show a tangle-gram of four trees. The leftmost tree represents the chromosomal data, and the next tree the corresponding spA sequences with more than 70% coverage of the reference genome. When using the 70% coverage threshold, the clade denoted by the red line is removed from the dataset.
(PDF)

**S5 Fig. Test for different tree structures between the different spA alignments.** Here, we use Bayesian inference to test for a signal of different trees between the different spA alignments used. To do so, we ran a coalescent with reassortment analyses using the three spA alignments as segments. We used an exponential with a mean of 0.01 as a prior on the reassortment rates. We used the coalescent with reassortment over plasmid transfer, as there is no main segment for this analysis. We then show the expected counts for the reassortment event under the prior compared to the inferred counts. As shown, the data between the three spA alignments increases the weight of zero events, showing an absence of signal for the different alignments to code for different evolutionary histories
(PDF)

**S6 Fig. Overview of spBC detection in the *S. sonnei* data.** Here, we show the approach for the spB and spC alignment. Panels A) and D) show the percentage cover of the reference plasmid for spB and spC, respectively. Panels B) and E) show the number of SNPs detected in each dataset for isolates with $\geq$90 of the plasmid reference with different core SNP filter thresholds. Panels C) and F) show the consistency index (CI) at different core SNP thresholds.
(PDF)

**S7 Fig. Inferred number of times at which plasmids jumped between bacterial lineages.** Here, we show the posterior distribution of how often a pINV (the virulence plasmid), spA, spB, and spC on the x-axis moved between *S. sonnei* lineages on the y-axis.
(PDF)

**S8 Fig. Comparison of the plasmid transfer rate between spA, spB, and spC for three separate analyses that use different parts of the spA plasmid**. Here, we compare the posterior distribution of the plasmid transfer rate when using different parts of the spA plasmid for inference. Each violin plot is created from a different analysis using either the entire spA plasmid, the combination of four AMR genes from *sul2* to *tetA*, the three AMR genes *sul2, strA,strB*, and the flanking region of ~100 bases.
(PDF)

**S9 Fig. Comparison of the plasmid transfer rate when using different parts of the spA plasmid.** Here, we compare the posterior distribution of the plasmid transfer rate, focusing only on spA and comparing the use of different parts of the spA plasmid for inference. Each violin plot is created from a different analysis using either the entire spA plasmid, the combination of four AMR genes from *sul2* to *tetA*, the three AMR genes *sul2, strA,strB*, and the flanking region of ~100 bases.
(PDF)

**S10 Fig. Rate at which the different plasmids are being lost by *S. sonnei* lineages.** We compute the rate at which plasmids are being lost as the number of events where a plasmid was lost divided by the tree length of that plasmid. We assume a plasmid was lost on the edges where the parent node carried a plasmid, while the child node did not.
(PDF)

**S11 Fig. Average rate estimates of plasmid gain and loss between observed and randomized data.** Here, we model the presence/absence of a plasmid at the tips using a continuous-time Markov chain. For each plasmid, we use once the actual observed presence/absence data and once a permutated tip to plasmid assignment. If the plasmid were always present, but is randomly lost during the sampling procedure, such as in culture, we would expect no difference between the gain and loss rates of plasmid for the true and permutated data. If there is substantial inherited structure in the presence or absence of plasmids, we would expect to estimate substantially lower rates in the true compared to the randomized tip to plasmid presence assignment.
(PDF)

**S12 Fig. Estimated rates of gain and loss of plasmids when modelling the presence and absence of each plasmid.** We infer the rates of gain and loss of pINV, spA, spB, and spC using a discrete trait model where we model the presence and absence of each plasmid as a continuous-time Markov chain. The rates show the posterior estimates, and the dots and numeric values show the median estimate.
(PDF)

**S13 Fig. Rates of evolution estimated for plasmid when modeling and when not modeling the joint history with the chromosomal DNA** We simulated 50 phylogenetic networks under the coalescent with plasmid transfer with three plasmids sampled over five years. We assume that the chromosome and the three plasmids evolved at a rate of $5 \times 10^{-4}$ subs/site/unit time. The chromosome has an SNP alignment length of 8000bp, while the three plasmids had SNP alignments of 200bp, 100bp, and 50bp, respectively. These settings will produce approximately the same number of SNPs per unit of time as a chromosome of 4.8 Mbp evolving at a rate of $8 \times 10^{-7}$ subs/site unit time. On the y-axis, we show the inferred evolutionary rates with the error bars denoting the 95% HPD and the point denoting the mean estimates. The x-axis is the number of variable sites in the alignment.
(PDF)

**S14 Fig. Comparison of the clock rates of chromosomal DNA and plasmid DNA.** Here, we compare the posterior distribution of the number of plasmid transfer events when using different parts of the spA plasmid for inference. Each violin plot is created from a different analysis using either the entire spA plasmid, the combination of four AMR genes from *sul2* to *tetA*, the three AMR genes *sul2, strA,strB*, and the flanking region of ~100 bases.
(PDF)

**S15 Fig. Estimate of the coefficient of variation when using a relaxed clock model to reconstruct the evolutionary history of the *S. sonnei* chromosome.** Here, we inferred the evolutionary history of *S. sonnei* using 400 random isolates of the chromosome sequences using a log-normal relaxed clock model in BEAST2 and a constant coalescent prior. We then simulate an alignment using the mean clock rates on top of a random tree in the posterior and a strict clock model. We next re-inferred the evolutionary history in the case where we know that the true clock model was a strict clock model and compared the coefficient of variation between the true data and the data simulated under a strict clock model. The true data shows some signal for rate variation across, however, the intervals are slightly overlapping, indicating that there is not substantial rate variation.
(PDF)

**S16 Fig. Overview of detection of MDR plasmid pKSR100 in the *S. sonnei* and *S. flexneri* datasets.** Here, we show the approach for the MDR plasmid pKSR100 alignments. Panel A) shows the percentage cover of the reference pKSR100

plasmid for *S. sonnei* and *S. flexneri*. Panel B) shows the number of SNPs detected in each dataset for isolates with four different thresholds for the coverage of the plasmid reference with different core SNP filter thresholds. Panels C) show the consistency index (CI) of four different thresholds for the coverage of the plasmid reference with different core SNP filter thresholds for three datasets.

(PDF)

**S17 Fig. Transmission of pKSR100 between *S. sonnei* and *S. flexneri*.** MCC network of *S. sonnei* and *S. flexneri* samples with the embedding of the pKSR100 plasmid tree **A**. The text denotes the posterior support values for plasmid transfer events. **B** Plasmid tree of pKSR100 with the host species *S. sonnei* or *S. flexneri* mapped onto the tree. The different colors of the tips show clusters of sequences that are the result of separate introductions of the MDR plasmid. MCC: maximum clade credibility. MDR: multidrug resistance

(PDF)

**S18 Fig. Comparison of network height, length, and plasmid transfer events between sampled and simulated networks.** To validate the implementation of CoalPT, we simulated networks under the CoalPT model, once with 5 plasmids and once with 10 plasmids. We then sampled phylogenetic networks under our implementation of the CoalPT model in BEAST2 under the prior (i.e., without any sequence information). As shown here, the summary statistics between networks simulated and sampled (using MCMC) under CoalPT match.

(PDF)

**S19 Fig. Schematic description of the Add/remove operator.** The add remove operator selects a random edge to add a plasmid transfer event to. On that edge, a random time is selected and a random plasmid is 'transferred'. We then sample the time to the next coalescent event under the constant coalescent, which becomes the reattachment time. Lastly, we select a random edge to reattach the plasmid edge to.

(PDF)

**S20 Fig. Inferred effective population sizes from simulated data.** To test the performance of the coalescent with plasmid transfer, we simulated 100 networks in a well-calibrated simulated study. The effective population sizes were sampled from a Lognormal distribution with M=1.4844 and S=0.5. The plasmid transfer rates were sampled from a Lognormal distribution with M=-1.7344 and S=0.5. We then simulated genomic sequences for the core genome and 3 plasmids under the Jukes-Cantor Model. Last, we inferred the phylogenetic network, effective population sizes, and plasmid transfer rates from these sequences using the above lognormal distributions as priors on the Ne and plasmid transfer rates. Here, we show the inferred Ne sizes (y-axis) compared to simulated Ne (x-axis). The point denotes the median estimate and the error bars the lower 95% highest posterior density interval.

(PDF)

**S21 Fig. Inferred plasmid transfer rates from simulated data.** Here, we show the inferred plasmid transfer rates(y-axis) compared to the true/simulated rates on the x-axis. These estimates are from the same analyses as the ones in S20 Fig. The point denotes the median estimate and the error bars the lower 95% highest posterior density interval.

(PDF)

### Acknowledgments

The Microbiological Diagnostic Unit Public Health Laboratory is supported by the State Government of Victoria, Australia.

### Author contributions

**Conceptualization:** Nicola Müller, Trevor Bedford, Sebastián Duchêne, Danielle J. Ingle.

**Data curation:** Nicola Müller, Danielle J. Ingle.

**Formal analysis:** Nicola Müller, Ryan R. Wick, Sebastián Duchêne, Danielle J. Ingle.

**Funding acquisition:** Trevor Bedford, Benjamin P. Howden, Danielle J. Ingle.

**Investigation:** Nicola Müller, Ryan R. Wick, Louise M. Judd, Sebastián Duchêne, Danielle J. Ingle.

**Methodology:** Nicola Müller, Sebastián Duchêne.

**Resources:** Benjamin P. Howden.

**Software:** Nicola Müller.

**Validation:** Nicola Müller.

**Visualization:** Nicola Müller, Danielle J. Ingle.

**Writing – original draft:** Nicola Müller, Sebastián Duchêne, Danielle J. Ingle.

**Writing – review & editing:** Nicola Müller, Ryan R. Wick, Louise M. Judd, Deborah A. Williamson, Trevor Bedford, Benjamin P. Howden, Sebastián Duchêne, Danielle J. Ingle.

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
