## [Decision Letter · Decision Letter 0]

22 Jun 2025

PPATHOGENS-D-25-00988

Quantifying plasmid movement in drug-resistant *Shigella* species using phylodynamic inference

PLOS Pathogens

Dear Dr. Müller,

Thank you for submitting your manuscript to PLOS Pathogens. After careful consideration, we feel that it has merit but does not fully meet PLOS Pathogens's publication criteria as it currently stands. Therefore, we invite you to submit a revised version of the manuscript that addresses the points raised during the review process.

Please submit your revised manuscript within 30 days Aug 21 2025 11:59PM. If you will need more time than this to complete your revisions, please reply to this message or contact the journal office at plospathogens@plos.org. Please include the following items when submitting your revised manuscript:

We look forward to receiving your revised manuscript.

Kind regards,

D. Ashley Robinson, Ph.D.

Academic Editor

PLOS Pathogens

Debra Bessen

Section Editor

PLOS Pathogens

Sumita Bhaduri-McIntosh

Editor-in-Chief

PLOS Pathogens

orcid.org/0000-0003-2946-9497

Michael Malim

Editor-in-Chief

PLOS Pathogens

orcid.org/0000-0002-7699-2064

**Additional Editor Comments:**

Please respond to each comment from the Reviewers. Note that Reviewer 3's comments are sent as a separate attachment.

**Journal Requirements:**

At this stage, the following Authors/Authors require contributions: Nicola Müller, Ryan R Wick, Louise M Judd, Deborah A Williamson, Trevor Bedford, Benjamin P Howden, Sebastián Duchêne, and Danielle J Ingle. Please ensure that the full contributions of each author are acknowledged in the "Add/Edit/Remove Authors" section of our submission form.

3) Your manuscript is missing the following sections: Methods.  Please ensure all required sections are present and in the correct order. Make sure section heading levels are clearly indicated in the manuscript text, and limit sub-sections to 3 heading levels. An outline of the required sections can be consulted in our submission guidelines here:

https://journals.plos.org/plospathogens/s/submission-guidelines#loc-parts-of-a-submission

5) We notice that your supplementary Figures are included in the manuscript file. Please remove them and upload them with the file type 'Supporting Information'. Please ensure that each Supporting Information file has a legend listed in the manuscript after the references list.

Potential Copyright Issues:

i) Figure 1. Please confirm whether you drew the images / clip-art within the figure panels by hand. If you did not draw the images, please provide (a) a link to the source of the images or icons and their license / terms of use; or (b) written permission from the copyright holder to publish the images or icons under our CC BY 4.0 license. Alternatively, you may replace the images with open source alternatives. See these open source resources you may use to replace images / clip-art:

7) Please ensure that the funders and grant numbers match between the Financial Disclosure field and the Funding Information tab in your submission form. Note that the funders must be provided in the same order in both places as well. Currently, the Financial Disclosure states there was no funding received.

**Reviewers' Comments:**

Reviewer's Responses to Questions

**Part I - Summary**

Reviewer #1: The authors present a new Bayesian method for better understanding plasmid dynamics, which is of general interest as antimicrobial resistance (AMR) poses a major global health threat, largely because AMR genes can spread between bacteria via mobile elements like plasmids. Tracking how plasmids move and spread resistance has been difficult with current computational tools. In response, this study introduces a new Bayesian phylogenetic network method that models the co-evolution of chromosomal and plasmid DNA, enabling researchers to reconstruct and quantify plasmid movement between bacterial lineages over time while accounting for uncertainty.

Applying this approach to a five-year dataset of Shigella, the authors analyzed the transfer rates of plasmids with different resistance and virulence profiles. This method offers valuable insights into how AMR-carrying plasmids spread and persist in bacterial populations, potentially improving strategies to combat antimicrobial resistance. I have a few minor comments that should be addressed.

Reviewer #2: Muller and co-authors presented here a new phylodynamic model named CoalPT, to investigate the evolutionary dynamics of AMR-carrying plasmids in Shigella sonnei and flexneri, also starting from short-read data. This model is based on coalescent process with reassortment, parametrized to quantify the transfer and evolutionary rates of plasmids of epidemiological interest, adding novel insights about plasmid movement across species, which is usually less studied due to its complexity.

The model is explained exhaustively and the workflow of the analysis is solid. I only have a few concerns that might need clarification, which I am expressing as following.

Reviewer #3: (No Response)

**Part II – Major Issues: Key Experiments Required for Acceptance**

Reviewer #1: (No Response)

Reviewer #2: Line 217: Did the authors include a representative number of samples x year during the random sampling? This could affect the estimation of Ne and relating coalescent process e.g. coalescent rate over time

Line 139-145 The authors observed that the number of events and transfer rates was lower when using only single AMR genes compared to the AMR genes + the flanking regions or the whole plasmid. It would be expected that if genes are moving together e.g. transposons are prone to high transfer rates, being more plastic in terms of HGT compared to the whole plasmid transfer. How was the convergence of the model when using only AMR rather than the other two alignments? It would be helpful to confirm the authors's hypothesis about the absence of phylogenetic signal using dedicated tools e.g. Tempest

Line 227 The authors parametrize the model to account for different plasmid rates, and they found evidence of rapid expansion of a clade in some cases. Why did the authors not consider using alternative models other than strict e.g. fixed local when calibrating molecular clock, which allows lineages to evolve under different evolutionary rates?

Reviewer #3: (No Response)

**Part III – Minor Issues: Editorial and Data Presentation Modifications**

Reviewer #1: - "Methods" section title missing

- line: 333/4 : "In fact, it is the backward-in-time equivalent of one plasmid being

transferred between bacterial lineages at a time." I don't understand what this equivalence refers to, please clarify.

- Please clearly state and discuss the independence assumption on events (plasmid transfer, coalescent, sampling events)

- Please discuss what assumptions are made for the sampling process - very low sampling proportion for both chromosome and plasmids? How realistic are these here and how much may that affect results? This could be addressed via simulations, but a thorough discussion would also suffice.

- Fig 2 "Proportion of lineages carrying a plasmid between 2000 and 2020." How much does this result depend on sampling scheme? And how does it compare to occurrence data? Could the method also include occurrence data =?

- Fig 2 "The text denotes the posterior probability of plasmid transfer

events for events with a posterior support of over 0.5. " What text does this refer to? Please fix and clarify.

- Fig 2 "Branches are colored to denote separate clades originating from individual jumps of plasmids into a bacterial clade. " Which colours does this refer to? Please fix and clarify.

- p.8 Please explain the choice of thresholds for the input alignments (80% plasmid coverage, 95% core conservation, 250 subsamples)

- Discussion: Please state clearly if applicable to plasmid movements between completely different bacteria. Even in a structured model this seems difficult if a joint alignment is required.

Reviewer #2: In Fig.2, how do the authors explain the scattered distribution of the plasmid on terminal tips? Would it be possible that those plasmids are prone to be lost by the strains when cultured, rather than actual loss for other reasons e.g. environmental selective pressures? The authors should discuss this potential issue

Line 474 Core-SNP-filter github reference can be replaced with the publicated paper https://doi.org/10.1099/mgen.0.001346

Fig. S6, S7, S9, S11 It would be easy to visualize the differences if the plots were shown using only one row

Fig S3,S5 please increase label size

Reviewer #3: (No Response)

PLOS authors have the option to publish the peer review history of their article (what does this mean?). If published, this will include your full peer review and any attached files.

Reviewer #1: No

Reviewer #2: No

Reviewer #3: No

**Figure resubmission:**
---

## [Editor Report · Decision Letter 1]

13 Oct 2025

Dear Prof. Müller,

We are pleased to inform you that your manuscript 'Quantifying plasmid movement in drug-resistant *Shigella* species using phylodynamic inference' has been provisionally accepted for publication in PLOS Pathogens.

Best regards,

D. Ashley Robinson, Ph.D.

Academic Editor

PLOS Pathogens

Debra Bessen

Section Editor

PLOS Pathogens

Sumita Bhaduri-McIntosh

Editor-in-Chief

PLOS Pathogens

orcid.org/0000-0003-2946-9497

Michael Malim

Editor-in-Chief

PLOS Pathogens

orcid.org/0000-0002-7699-2064
---

## [Editor Report · Acceptance letter]

Dear Prof. Müller,

We are delighted to inform you that your manuscript, "Quantifying plasmid movement in drug-resistant *Shigella* species using phylodynamic inference," has been formally accepted for publication in PLOS Pathogens.

Best regards,

Sumita Bhaduri-McIntosh

Editor-in-Chief

PLOS Pathogens

orcid.org/0000-0003-2946-9497

Michael Malim

Editor-in-Chief

PLOS Pathogens

orcid.org/0000-0002-7699-2064